# Integrative Analysis of the Transcriptome and Metabolome Reveals the Mechanism of Chinese Fir Seed Germination

Xiangteng Chen, Guangyu Zhao, Yanglong Li, Shumeng Wei, Yuhong Dong and Ruzhen Jiao *

Research Institute of Forestry, Chinese Academy of Forestry, Beijing 100091, China
* Correspondence: jiaorzh@caf.ac.cn

**Abstract:** Chinese fir (*Cunninghamia lanceolata* (Lamb.) Hook.) is an important plantation tree species in China, and seed germination is a key step in forest tree cultivation. To reveal the gene expression network and molecular mechanisms in the germination of Chinese fir seeds, physiological indexes were measured and transcriptome and metabolome analyses were performed on Chinese fir seeds in four stages of germination (imbibition stage, preliminary stage, emergence stage, and germination stage). All six physiological indicators had significant differences at different developmental stages. In transcriptome and metabolome analysis, we identified a large number of differentially expressed genes (DEGs) and differentially accumulated metabolites (DAMs). Gene Ontology (GO) analysis showed a large number of DEGs associated with cell growth, and Kyoto Encyclopedia of Genes and Genomes (KEGG) enrichment analysis showed that DEGs were significantly enriched in the flavonoid biosynthesis, phenylpropanoid biosynthesis, and plant hormone signal transduction pathways. The KEGG enrichment results of DAMs were similar to those of DEGs. The joint analysis of DEGs and DAMs indicated that flavonoid biosynthesis and phenylpropanoid biosynthesis were the key pathways of Chinese fir seed germination. Our study revealed a number of key genes and key metabolites, laying the foundation for further studies on the gene regulatory network of Chinese fir seed germination.

**Keywords:** Chinese fir; seed germination; transcriptome; metabolome; mechanism

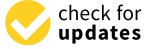



## 1. Introduction

Chinese fir (*Cunninghamia lanceolata*) is an important plantation tree species in south China and widely distributed in many southern provinces, and it has the advantages of a wide cultivation range, a fast growth rate, and a high economic value [1–3]. Previous studies on Chinese fir have focused on resource investigation, cultivation, and pests and diseases [4–6]. However, there are fewer studies on seed germination, and the existing studies are still mainly about the use of physical, chemical, and biological methods to enhance the germination rate of seeds, while studies on the seed germination process have not been reported. Chinese fir seed is characterized by a low germination rate and low yield, and it is crucial to study the mechanism of Chinese fir seed germination for the development of Chinese fir.

Seed germination is an extremely complex process that is influenced by multiple environmental factors [7]. Previous studies have focused on physiological and morphological changes in seed germination [8], but the molecular mechanism of germination has been less studied. Seed germination requires large amounts of energy and nutrients, and the seed cannot photosynthesize during the germination stage. At this time, genes associated with this process are activated and regulated to provide the material needed for seed germination. Studies have proven that seed germination is strictly regulated by plant hormones; ABA and GA are the most important plant hormones in seed germination, and both are antagonistic to seed germination [9,10]. It has been shown that flavonoids have an important role in seed germination [11–13]. Seed germination is regulated by a complex network

of gene regulation, signal transduction, and metabolic changes [14]. Transcriptomic and metabolomic technologies can identify and recognize genetic and metabolic changes during biological development and analyze the regulatory mechanisms of plants from two aspects. In recent years, more studies have been conducted on the metabolic and transcriptional changes involved in the seed germination of agricultural and cash crops [15,16].

In forestry production, the germination of tree seeds is the first and most important step, directly related to the subsequent seedling growth and timber. However, changes at the level of transcription and metabolism involved in the germination process of Chinese fir seeds have not been reported. To investigate the potential mechanisms of Chinese fir seed germination, physiological indicators, transcription, and metabolic analyses were performed on Chinese fir seeds at four stages in germination. The changes of physiological indicators during the germination of Chinese fir seed were studied, and the key regulators in the pathways of phenylpropanoid biosynthesis and flavonoid biosynthesis were identified and their regulatory modes were analyzed. This study provided some theoretical basis for the breeding of Chinese fir.

## 2. Materials and Methods

### 2.1. Experiment Materials and Settings

The Chinese fir seeds harvested in the Hongya State Forestry (Meishan, China) were selected as experimental materials, and the seeds were of the same size and full of seeds. These seeds needed to be washed with 0.5% $KMnO_4$ for 10 min and then washed 5 times with distilled water before germination. After soaking the seeds in distilled water for 24 h and absorbing water from the surface of the seeds with filter paper, these Chinese fir seeds were placed in germination boxes with 2 layers of gauze. The germination boxes were placed in a photostat incubator with a photoperiod of 16 h at 25 °C, a dark cycle of 8 h at 20 °C, and a light intensity of 284 µmol/($m^2 \cdot s$). During the experiment, we observed the seeds germination every day and took timely samples.

Here, we divided seed germination of Chinese fir into four stages: imbibition stage (S1), preliminary stage (S2), emergence stage (S3), and germination stage (S4) (Figure 1). In addition, the four germination stages were sampled on the first, third, fifth, and eighth days of the germination experiment. In the study, thirty Chinese fir seeds were collected from each stage, and every five Chinese fir seeds formed a composite sample with three biological replicates. The seed samples were collected and partially placed at −80 °C for transcriptome and metabolome analysis and partially dried for the analysis of physiological indicators.

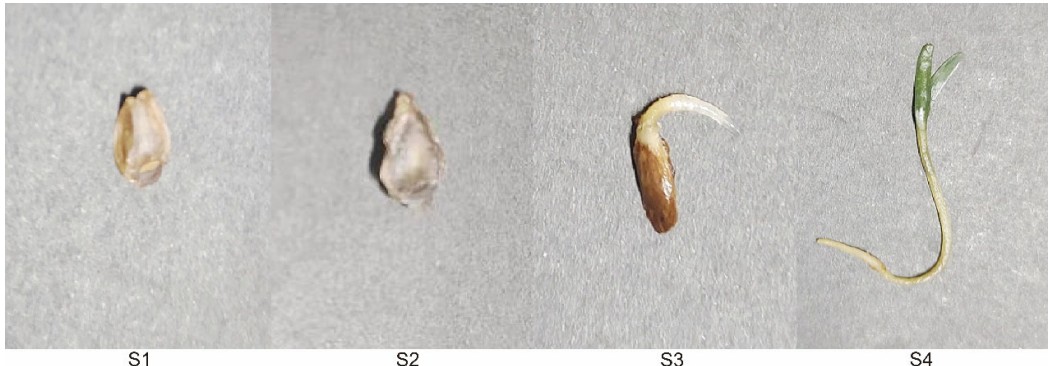

**Figure 1.** Chinese fir seeds in the four germination stages (imbibition stage (S1), preliminary stage (S2), emergence stage (S3), and germination stage (S4)).

### 2.2. Determination of SS, SP, MDA, Pro, SOD, and POD

After seed germination, the seeds from four stages were collected and tested for different physiological indices, three times per stage. The anthrone colorimetric method was used to determine soluble sugar (SS) content [17]. Soluble protein (SP) content was measured as described by Howarth [18]. Malondialdehyde (MDA) content was measured

according to the methods of Velikova [19], and proline (Pro) content was measured as described by Bates [20]. The activity of superoxide dismutase (SOD) was measured using the o-Benzenetriol method [21], and the Guaiacol colorimetric assay was used to determine peroxidase (POD) activity [22].

### 2.3. Total RNA Extraction and Transcriptome Sequencing

Approximately 0.1 g of samples (12 samples; 3 replicates of each germination stage) were placed in sterile centrifuge tubes, and the RNA of the seed was extracted with RNAisoPlus extraction reagent (Tiangen, China). We constructed sequencing libraries with the RNA Library Prep Kit for Illumina®® (NEB, San Diego, CA, USA), and RNA needed to be purified in the AMPure XP system (Beckman Coulter, Beverly, MA, USA). Then library quality was assessed on the Agilent Bioanalyzer 2100 system. Finally, the constructed libraries were uploaded and sequenced on the NovaSeq 6000 platform. The sequenced reads were compared with the Unigene library using Bowtie, and the expression levels were estimated based on the results of the comparison in combination with RSEM. The FPKM values were used to express the expression abundance of the corresponding Unigene. Differentially expressed genes (DEGs) selection between seed development stages was based on DESeq2 software (v1.6.3), and screening criteria were a False Discovery Rate (FDR) < 0.01 and a Fold Change (FC) $\geq$ 2.

The DEGs obtained for each comparison group will be annotated in the databases (NR, Swiss-Prot, KEGG, COG, KOG, GO, and Pfam). Gene Ontology (GO) was a standard biological annotation database that established the relationship between genes and their product functions. As the material for this experiment was reference-free, GO functional annotation of genes from this species was required prior to enrichment analysis. The annotated GO was then used as the background gene set for GO enrichment analysis. The DEGs of different seed development stages were analyzed for GO enrichment using the topGO R packages (v2.28.0). The Kyoto Encyclopedia of Genes and Genomes (KEGG) [23] was a database resource that could be used to determine the function of genes. These DEGs were analyzed for KEGG enrichment, and significant enrichment pathways were counted between different comparison groups. In addition, KOBAS [24] software (v3.0) was used to examine the statistical enrichment of DEGs in KEGG pathways.

### 2.4. Quantitative Real-Time PCR (qRT-PCR) Analysis

The six genes were used for the qRT-PCR analysis (Supplementary Table S2). We extracted total RNA from samples with the Total Plant RNA Extraction Kit (Tiangen, Beijing, China), and detected RNA concentration and purity with NanoDrop®® ND-2000. The HiScript III 1st Strand cDNA Synthesis Kit (+gDNA wiper) (Vazyme, Nanjing, China) was used for cDNA reverse transcription, and the experiments were performed according to the product instructions. PCR reactions were performed by placing the 96-PCR plate on a real-time PCR instrument, and the PCR reaction procedure was 95 °C for 10 min and 40 PCR cycles (95 °C for 15 s and 60 °C for 60 s). After the amplification reaction, to establish the melting curve of the PCR products, the products were analyzed for specificity at 60–99 °C. Finally, the target gene and the internal reference of each sample were subjected to real-time PCR reactions separately, and three replicate wells were tested for each sample, and the data were analyzed using the $2^{-\Delta\Delta CT}$ method.

### 2.5. Metabolite Extraction and Analysis

The obtained samples (12 samples, 3 replicates of each germination stage) were weighed at 50 mg and extracted with 1000 μL of extraction solution containing an internal standard (methanol:acetonitrile = 2:1, internal standard concentration: 2 mg/L). This was followed by the addition of an appropriate amount of extract for re-solubilization and testing on the machine. The 12 samples were analyzed based on the LC-QTOF platform by a high-resolution mass spectrometer (Xevo G2-XS QTof).

The identified compounds were searched for classification and pathway information in the Kyoto Encyclopedia of Genes and Genomes (KEGG), Human Metabolome Database (HMDB) [25], and Lipid Metabolites and Pathways Strategy (LIPID MAPS) [26] databases. The method of combining the difference multiple, the "*p*" and the VIP-value of the OPLS-DA model was adopted to screen the differential metabolites (DAMs). The screening criteria were Fold Change (FC) > 1, "*p*" < 0.05, and Variable Importance in Projection (VIP) > 1. The DAMs of KEGG pathway enrichment significance were calculated using a hypergeometric distribution test [27].

### 2.6. Statistical Analysis of Data

Excel 2016 was used to pre-arrange the data, and SPSS 17.0 was used to analyze the physiological indexes of Chinese fir seeds and a regression analysis between qRT-PCR and RNA-seq results. Figures were generated by Origin 2018 (9.0).

## 3. Results

### 3.1. Physiological Changes of Chinese Fir Seeds

We measured six indicators of the seed germination process. These results showed that SS, SP, Pro content, and the activities of SOD and POD reached their maximums at mid-seed germination (S2 or S3), except for MDA. Soluble sugar (SS) and soluble protein (SP) contents were highest in S2. Malondialdehyde (MDA) and proline (Pro) content were significantly different in four stages; malondialdehyde (MDA) content reaches a maximum at S4 and a minimum at S3; and proline (Pro) content was highest in S1 and lowest in S4. The activities of superoxide dismutase (SOD) and peroxidase (POD) were highest in S2, and both showed an upward and then downward trend (Figure 2).

### 3.2. Transcription Analysis

Transcriptome sequencing was carried out on 12 samples, and the raw reads were filtered to give clean reads. The clean reads of each sample reached 5.90 Gb, the percentage of Q30 bases $\geq$ 93.25%, the percentage of Q20 bases $\geq$ 97.50%, and the GC content $\geq$ 43.91% (Supplementary Table S1). These results indicated that the sequencing quality was good enough for further analysis.

The criteria for screening differentially expressed genes (DEGs) were FDR < 0.01 and FC $\geq$ 2, and DEGs among different developmental stages of Chinese fir seeds were screened. The DEGs were 6888 (Up/Down, 3897/2991) between stage 1 and stage 2, the DEGs were 8454 (Up/Down, 4672/3782) between stage 1 and stage 3, and the DEGs were 9216 (Up/Down, 5304/3912) between stage 1 and stage 4. Compared to the other two groups (S2/S1, S3/S1), there were more genes in S4/S1 (Figure 3A and Supplementary Table S3). The Venn diagram showed that 2365 commonly upregulated DEGs and 2665 commonly downregulated DEGs were found in three comparison groups (Figure 3B).

GO (Gene Ontology) could classify the function of genes, dividing their functions into three categories: molecular function (MF), cellular component (CC), and biological process (BP). GO results showed that there were 4867 DEGs, 6030 DEGs, and 6524 DEGs in S2/S1, S3/S1, and S4/S1, respectively. The DEGs of S2/S1 were annotated in 51 GO terms, and the largest number of DEGs in the three categories (MF, BP, and CC) was binding (2413 genes), metabolic process (2177 genes), and membrane (1643 genes). The DEGs in S3/S1 were distributed among 53 GO terms; the largest number of DEGs in the three categories (MF, BP, and CC) was binding (3044 genes), metabolic process (2777 genes), and cell (2022 genes). The DEGs were included in 52 GO terms in S4/S1. The largest number of DEGs in the three categories (MF, BP, and CC) was binding (3302 genes), catalytic activity (2967 genes), and cell (2187 genes). In addition, we could find the most DEGs related to the cellular component (Figure 3C and Supplementary Figures S1 and S2).

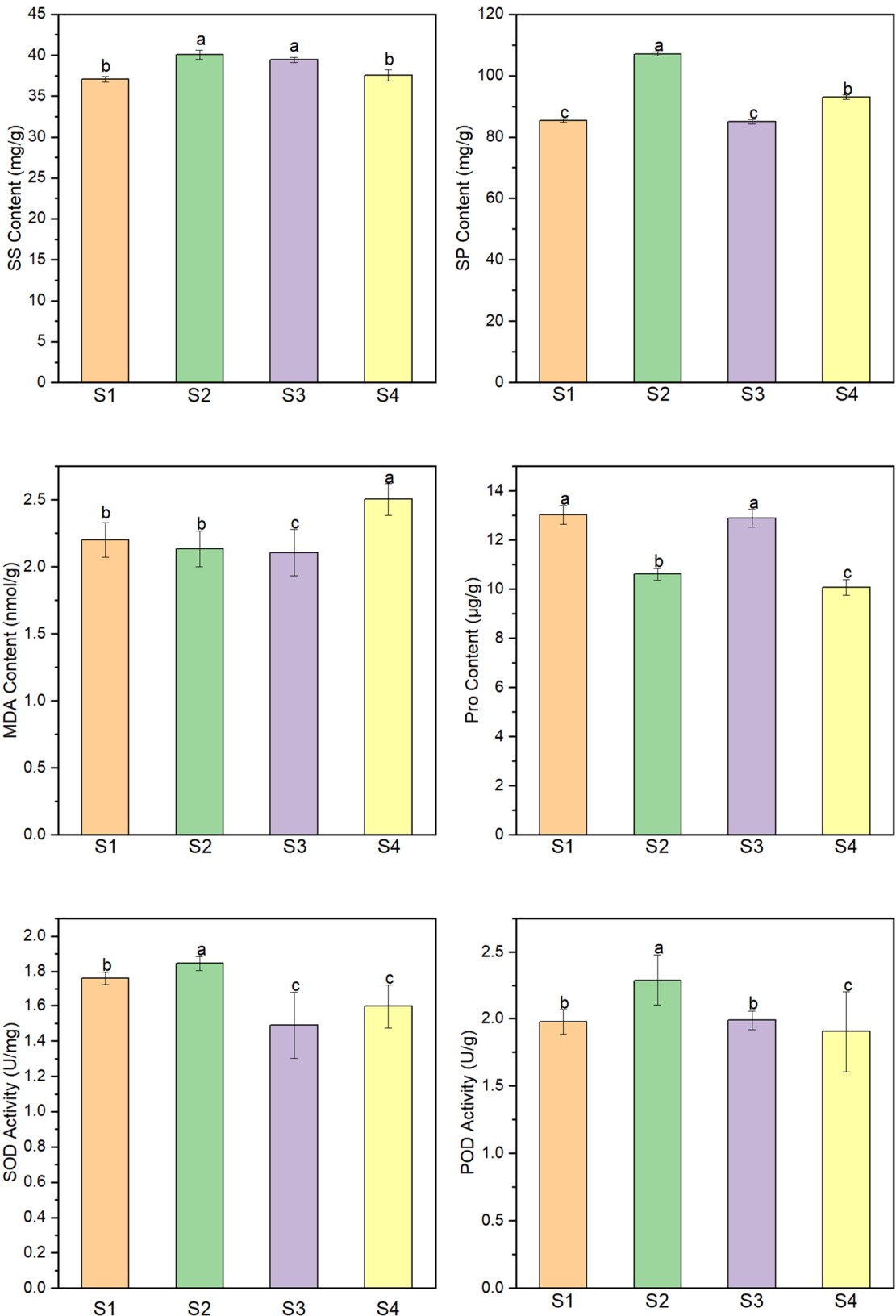

**Figure 2.** Physiological changes of Chinese fir seed germination, six indices (SS, SP, MDA, Pro, SOD, and POD). S1, S2, S3, and S4 represent the imbibition stage, the preliminary stage, the emergence stage, and the germination stage, respectively. Additionally, all the values are means ± standard errors, and different letters above the bars indicate significant differences.

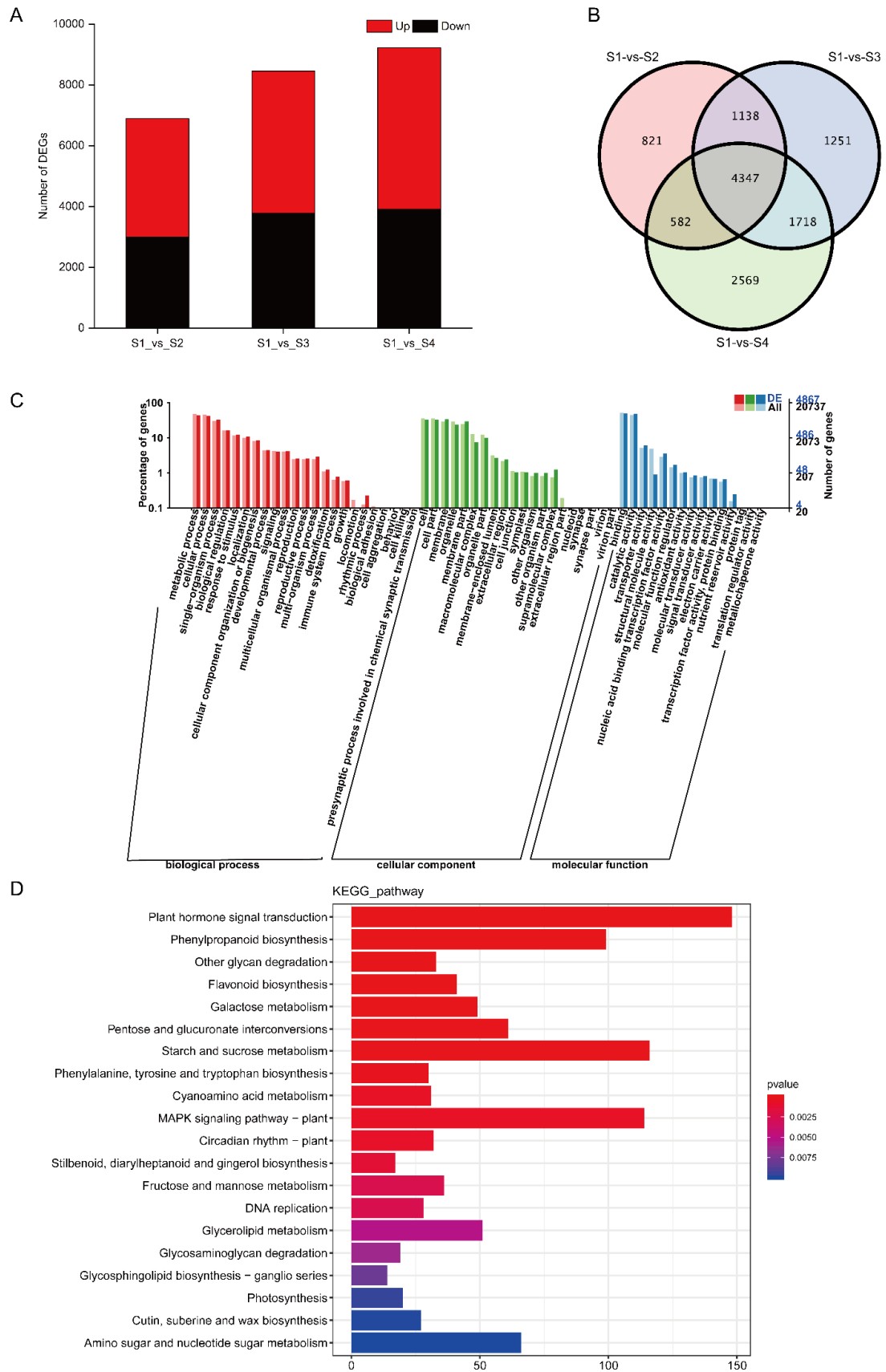

**Figure 3.** DEGs in four developmental stages of seeds. (**A**) Numbers of DEGs. (**B**) A Venn diagram of DEGs with S2/S1, S3/S1, and S4/S1. (**C**) GO classification of DEGs between S2 and S1. (**D**) KEGG pathways of DEGs between S2 and S1.

The KEGG (Kyoto Encyclopedia of Genes and Genomes) database was the most commonly used database for transcriptome analysis, enabling systematic analysis of gene function and metabolic pathways. In this study, we annotated and analyzed differentially expressed genes between developmental stages using the KEGG database. In S2/S1, the 2070 DEGs were annotated in 132 KEGG pathways, and there were 16 significantly enriched pathways (q < 0.05). For example, plant hormone signal transduction (48 DEGs, enrichment factor: 1.53); phenylpropanoid biosynthesis (99 DEGs, enrichment factor: 1.68); other glycan degradation (33 DEGs, enrichment factor: 2.22); flavonoid biosynthesis (41 DEGs, enrichment factor: 1.92); and galactose metabolism (49 DEGs, enrichment factor: 1.72). In S3/S1, the 2555 DEGs were annotated in 132 KEGG pathways, and there were 19 significantly enriched pathways (q < 0.05), including photosynthesis (36 DEGs, enrichment factor: 2.42); other glycan degradation (35 DEGs, enrichment factor: 1.91); circadian rhythm—plant (41 DEGs, enrichment factor: 1.77); porphyrin and chlorophyll metabolism (33 DEGs, enrichment factor: 1.67); and phenylpropanoid biosynthesis (95 DEGs, enrichment factor: 1.31). In S4/S1, the 2800 DEGs were annotated in 132 KEGG pathways, and 20 pathways were significantly enriched. For instance, plant hormone signal transduction (180 DEGs, enrichment factor: 1.38); phenylpropanoid biosynthesis (116 DEGs, enrichment factor: 1.46); other glycan degradation (39 DEGs, enrichment factor: 1.94); cutin, suberine, and wax biosynthesis (42 DEGs, enrichment factor: 1.76), and glycosphingolipid biosynthesis—ganglio series (22 DEGs, enrichment factor: 2.19). The results suggested that these pathways may be vital for the germination of Chinese fir seed (Figure 4 and Supplementary Figures S3 and S4).

Furthermore, we randomly selected six DEGs (c186613.graph_c0, c237436.graph_c0, c193003.graph_c0, c205540.graph_c0, c218337.graph_c0, and c222258.graph_c1) from S1, S2, S3, and S4 for qRT-PCR analysis to verify the reliability of the RNA-seq results. The results showed that RNA-seq analysis and qRT-PCR analysis were not significantly different and had a similar trend. It indicated that the RNA-seq data was reliable (Supplementary Figure S5 and Table S2).

*3.3. Metabolome Analysis*

For better coverage and detection of metabolites, we used two modes (positive ion mode and negative ion mode) in the LC-QTOF analysis.

This standard (VIP value > 1 and *p* < 0.05) was used to identify the differential metabolites (DAMs) between different developmental stages of Chinese fir seeds.

In positive ion mode, the DAMs were 632 (Up/Down, 330/302) between stage 1 and stage 2; the DAMs were 693 (Up/Down, 361/332) between stage 1 and stage 3; and the DAMs were 721 (Up/Down, 350/371) between stage 1 and stage 4. There were 488 common differential metabolites in the three comparison groups (Figure 5A,B and Supplementary Table S4). The first PC (PC1) explained 60.79% of the variation in the principal component analysis (PCA), and the second PC (PC2) was 31.18% (Figure 5C,D). PC1 and PC2 were 71.90% and 23.66% in negative ion mode (Figure 5D), and the DAMs were 275 (Up/Down, 197/78) between stage 1 and stage 2; the DAMs were 298 (Up/Down, 198/100) between stage 1 and stage 3; and 310 (Up/Down, 192/118) between stage 1 and stage 4. The common DAMs of the three comparison groups were 210 genes (Figure 5A,B and Supplementary Table S5). A cluster analysis of the DAMs of the samples showed significant differences in different developmental stages (Figure 5E,F).

We counted the 20 upregulated/downregulated DAMs with the largest fold change in positive and negative ion modes. The main DAMs were LysoPE(+), geldanamycin(+), mesuaxanthone B(+), malonylmalvin(+), Lys−TyrMe−OH(-), nystatin(-), glutathione(-), and acacetin(-) in four stages (Supplementary Tables S4 and S5).

The DAMs were annotated in the HMDB and KEGG databases. In the HMDB analysis of four stages, the results showed that DAMs mainly were carboxylic acids and derivatives, fatty acyls, benzene and substituted derivatives, glycerophospholipids, prenol lipids, and flavonoids.

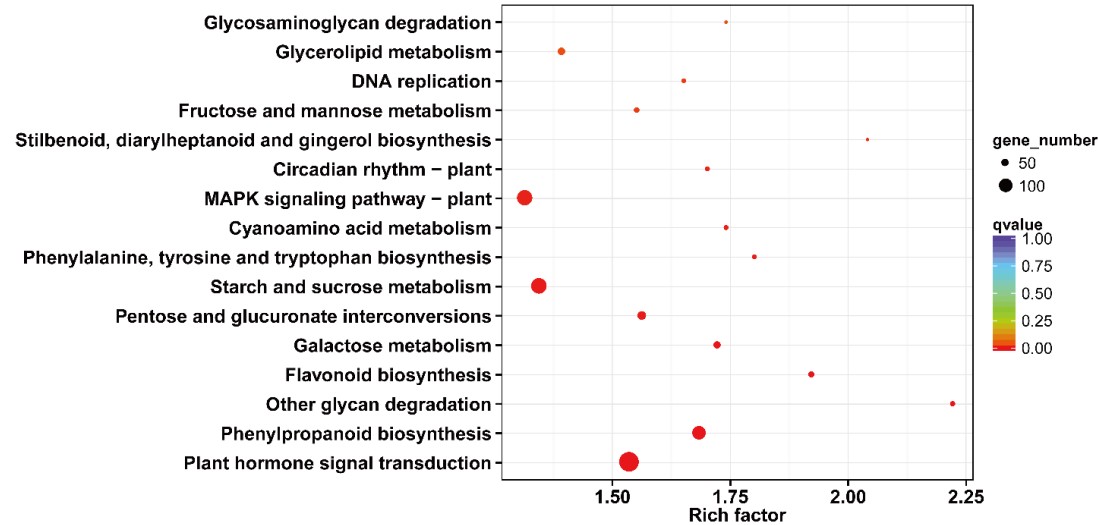

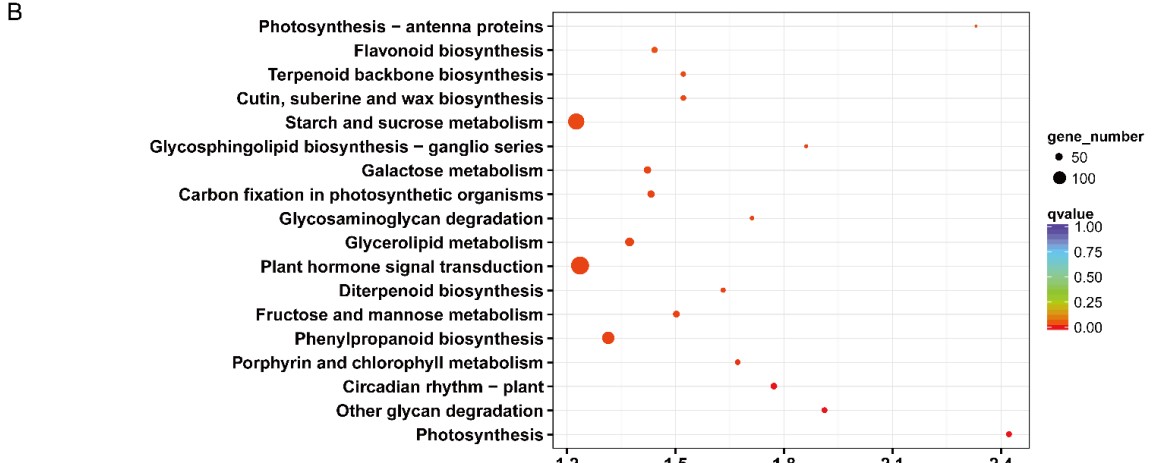

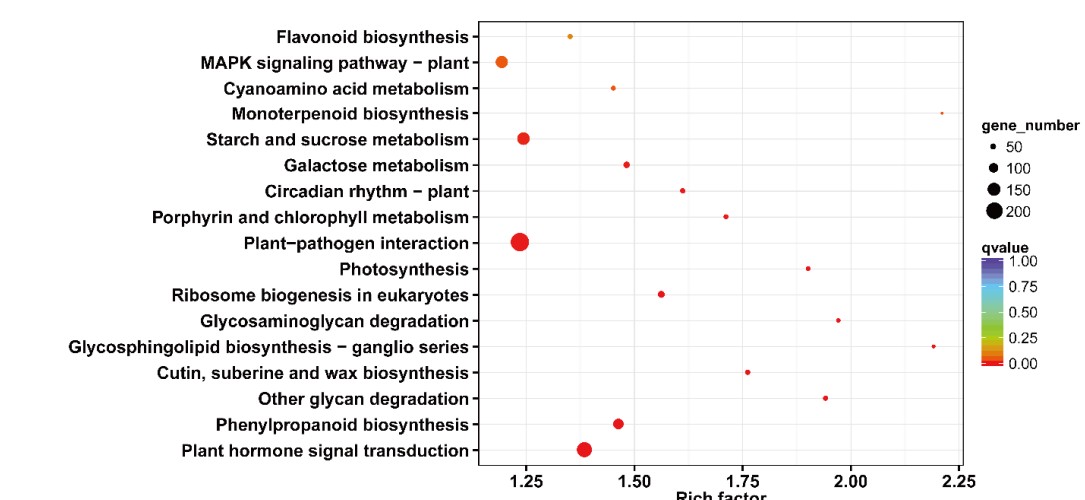

**Figure 4.** KEGG pathway enrichment of DEGs, (**A**) with S2/S1, (**B**) with S3/S1, and (**C**) with S4/S1. The rich factor indicates the ratio of the DEGs amount to the total amount of annotated genes in the pathway.

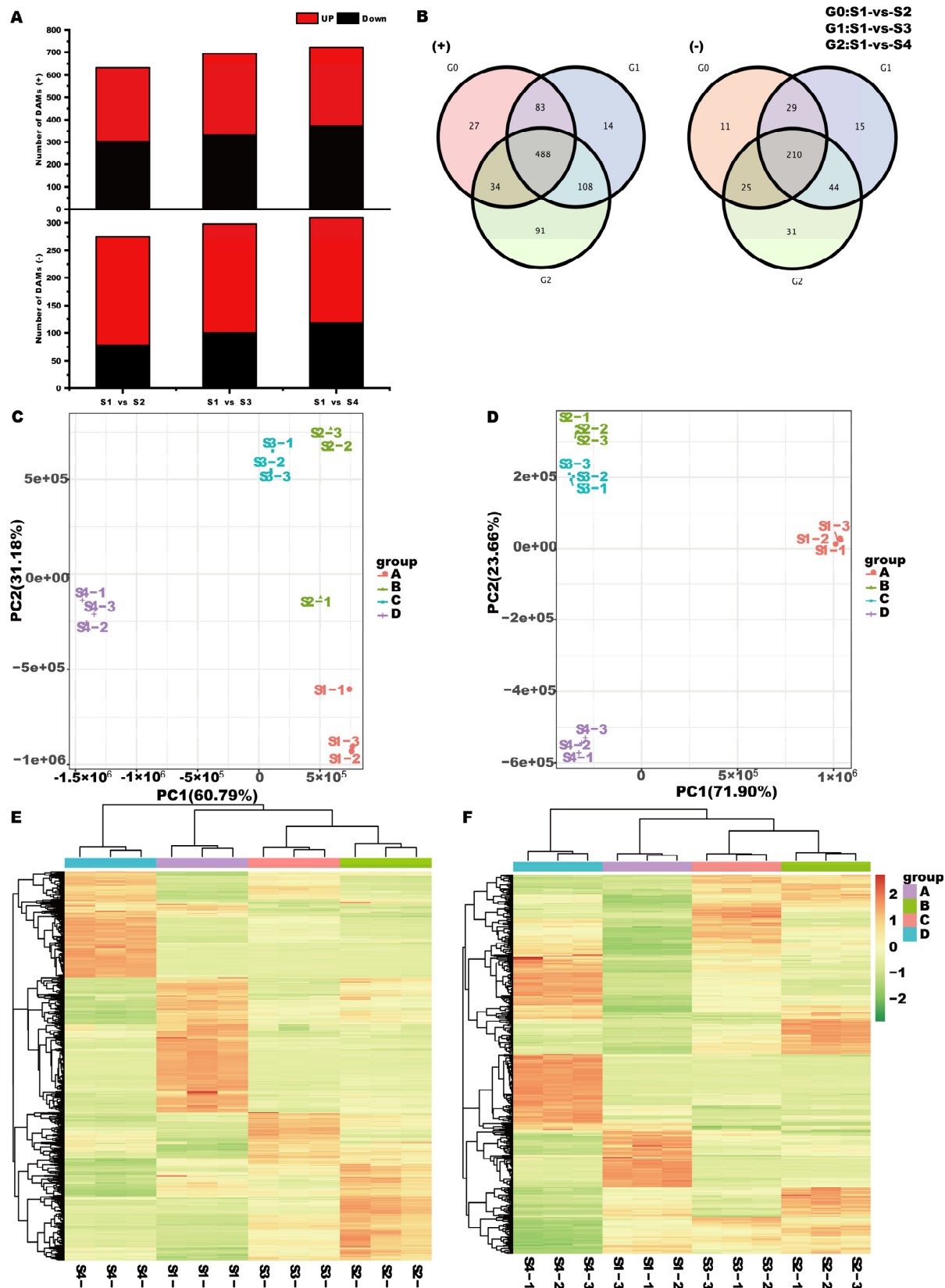

**Figure 5.** Metabolome analysis of Chinese fir seeds during germination. (**A**) DAMs of seeds. (**B**) A Venn diagram in positive ion mode and negative ion mode. (**C**) PCA of samples in positive ion mode. (**D**) PCA of samples in negative ion mode. (**E**) A heatmap of DAMs in positive ion mode. (**F**) A heatmap of DAMs in negative ion mode.

In the KEGG annotation results, the significantly enriched pathways were 'Flavonoid biosynthesis', 'Pyrimidine metabolism', 'Alanine, aspartate and glutamate metabolism', 'Flavone and flavonol biosynthesis', and 'Phenylalanine metabolism' in S2/S1 (Figure 6A); 'ABC transporters', 'Flavonoid biosynthesis', 'Phenylpropanoid biosynthesis', and 'Purine metabolism' in S3/S1 (Figure 6B); 'ABC transporters', 'Tryptophan metabolism', 'Flavonoid biosynthesis', 'Biosynthesis of amino acids', and 'Isoflavonoid biosynthesis' in S4/S1 (Figure 6C).

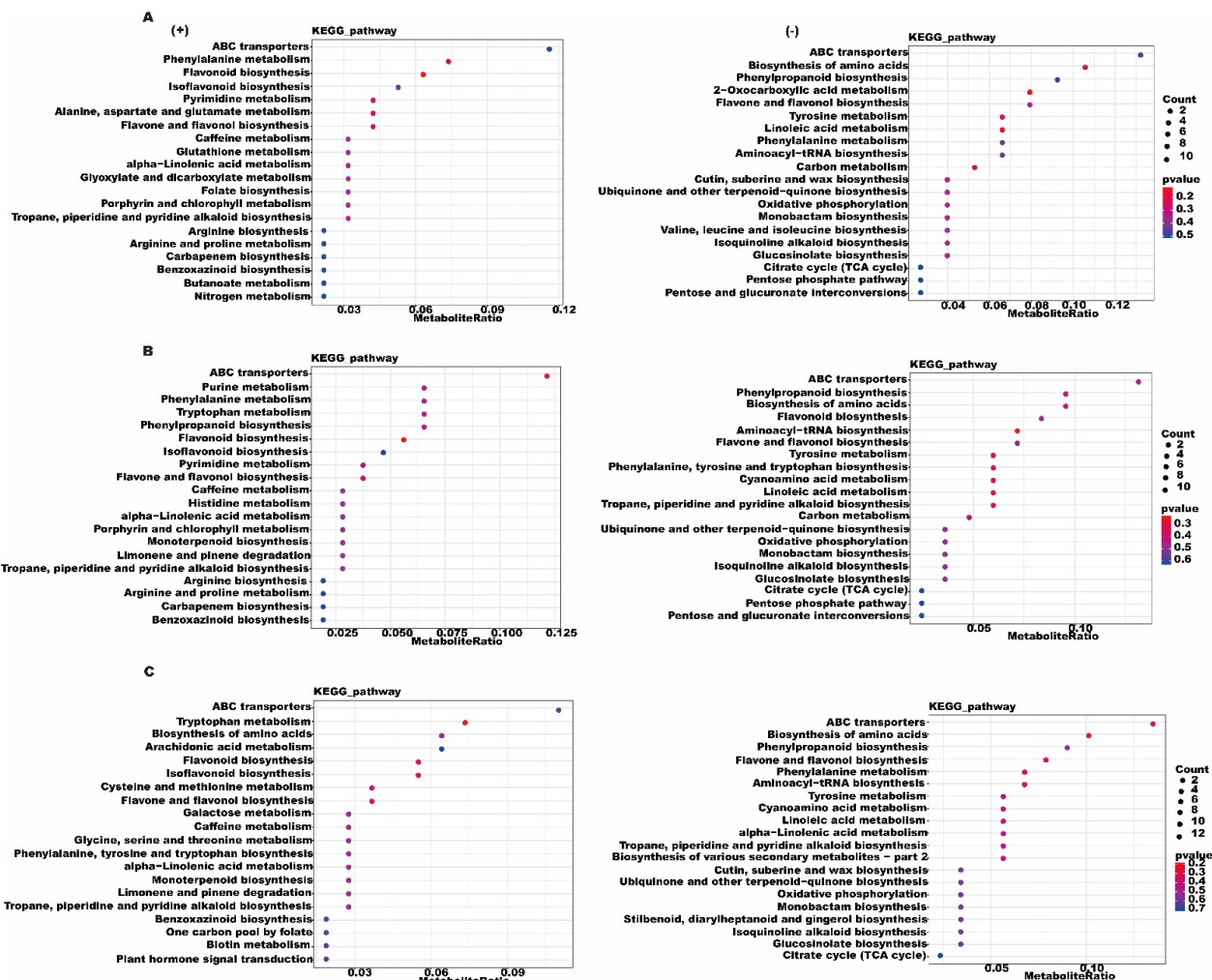

**Figure 6.** KEGG pathway enrichment of DAMs, (**A**) with S2/S1, (**B**) with S3/S1, and (**C**) with S4/S1. The left column represents the positive ion mode, and the right column represents the negative ion mode.

### 3.4. Integrated Metabolomic and Transcriptomic Analyzes

To further analyze the DEGs and DAMs, we annotated them in the KEGG database and obtained many joint pathways. There were 50, 55, and 59 joint pathways in three comparison groups, with the most significantly enriched pathways being 'Phenylpropanoid biosynthesis', 'Flavonoid biosynthesis', 'Pentose and glucuronate interconversions', 'Starch and sucrose metabolism', and 'Phenylalanine, tyrosine and tryptophan biosynthesis' in S2/S1 (Figure 7A); 'Photosynthesis', 'Phenylpropanoid biosynthesis', 'Diterpenoid biosynthesis', 'Plant hormone signal transduction', 'Carbon fixation in photosynthetic organisms', and 'Flavonoid biosynthesis' in S3/S1 (Figure 7B); 'Carbon metabolism', 'Phenylpropanoid biosynthesis', 'Diterpenoid biosynthesis', 'Flavonoid biosynthesis', 'Purine metabolism', and 'beta-Alanine metabolism' in S4/S1 (Figure 7C). We could find that the joint enriched pathways were 'Phenylpropanoid biosynthesis' and 'Flavonoid biosynthesis' in the

seed germination of Chinese fir. Therefore, these two pathways play an important role in seed germination.

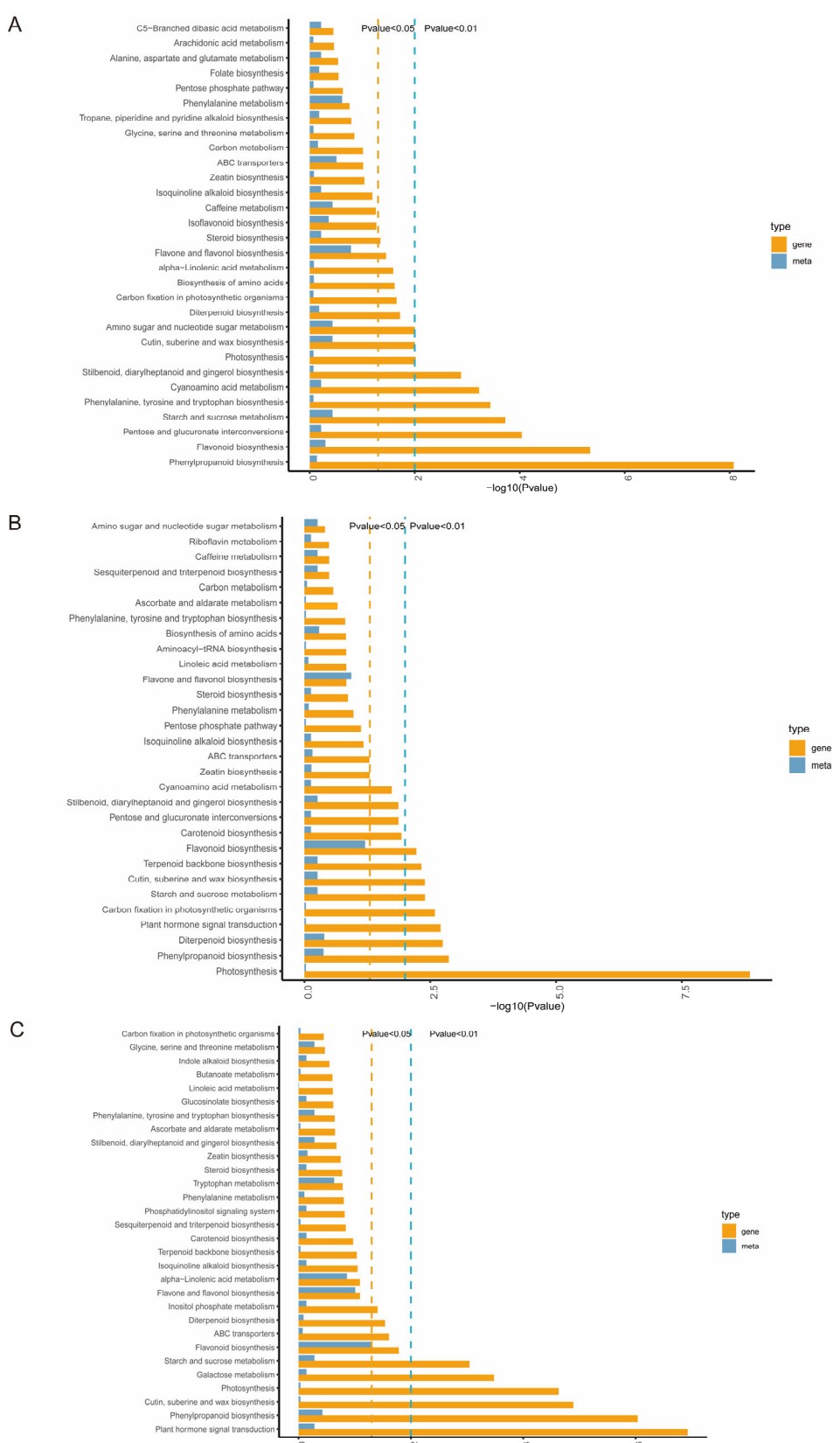

**Figure 7.** KEGG pathway enrichment of DEGs and DAMs, (**A**) with S2/S1, (**B**) with S3/S1, and (**C**) with S4/S1.

In order to explore the key pathways in the process of seed germination, we analyzed the two pathways in greater detail. In the phenylpropanoid biosynthesis pathway, most DEGs were upregulated, such as CYP73A, 4CL, COMT, CCR, and PER. Additionally, other DEGs were downregulated. The expression of CCR was increased in S2/S1(log$_2$FC, 4.96), S3/S1(log$_2$FC, 5.74), and S4/S1(log$_2$FC, 5.39). P-coumaroyl-CoA increased during seed germination in S2/S1(log$_2$FC, 2.38), S3/S1(log$_2$FC, 1.93), and S4/S1(log$_2$FC, 2.66). However, 5-Hydroxy-coruferaldehyde decreased during seed germination in S2/S1(log$_2$FC, −1.59), S3/S1(log$_2$FC, −1.31), and S4/S1(log$_2$FC, −1.96) (Figure 8A).

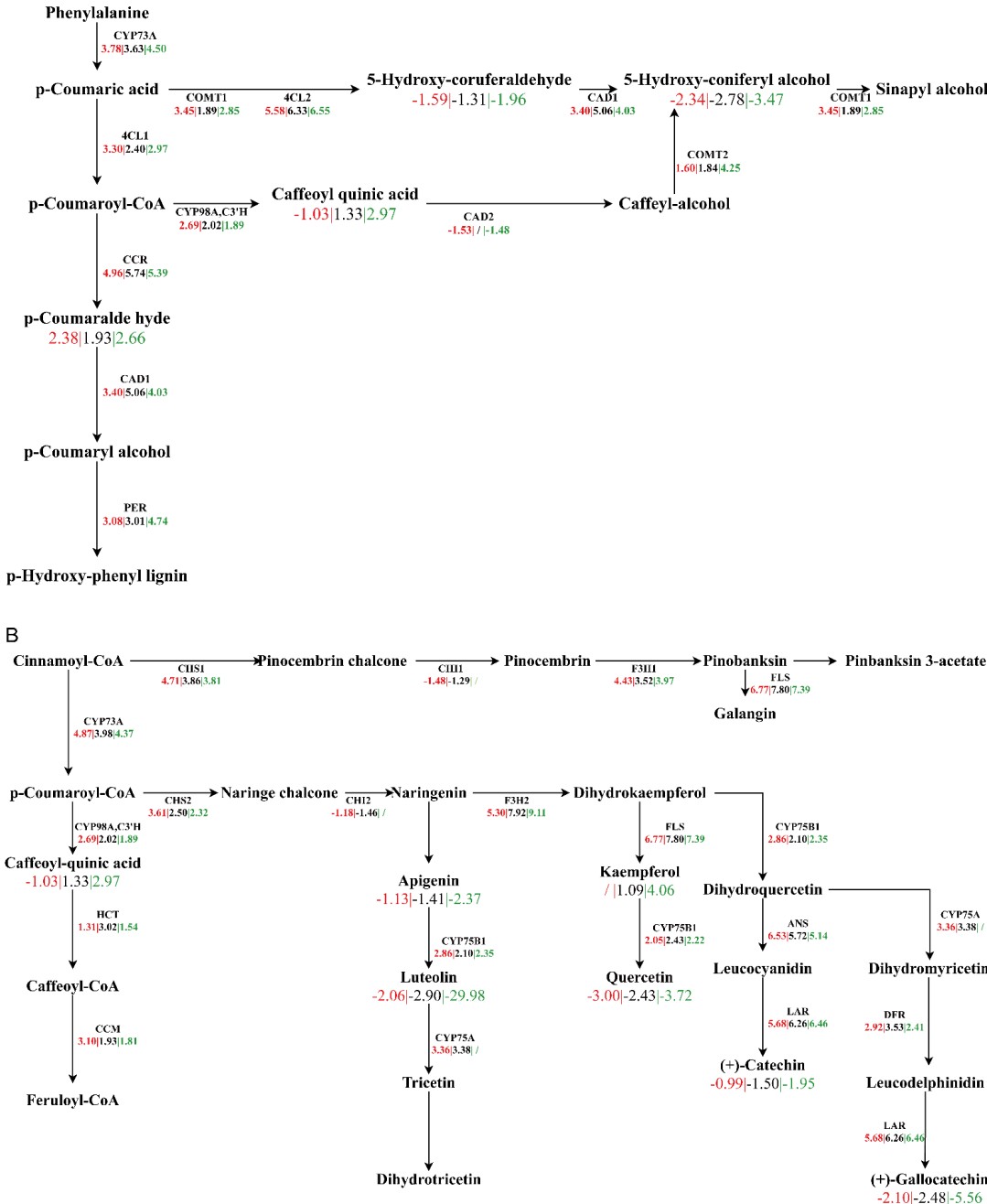

**Figure 8.** Integrated enrichment analysis of DAMs and DEGs related to phenylpropanoid biosynthesis (**A**) and flavonoid biosynthesis (**B**). At the location of the nodes are metabolites, and the substances on the line are genes; the metabolites and genes in italics were not detected; the red numbers represent the |log2fold change| of genes or metabolites for S2/S1, the black numbers represent S3/S1, and the green numbers represent S4/S1.

In the flavonoid biosynthesis pathway, many DEGs were upregulated. CHS, F3H, FLS, CCM, ANS, DFR, and LAR increased, but CHI decreased in S2/S1($\log_2$FC, $-1.48$) and S3/S1($\log_2$FC, $-1.29$). We found that kaempferol decreased in S2/S1 and increased in S3/S1, and S4/S1($\log_2$FC, 1.09; $\log_2$FC, 4.06). Additionally, caffeoyl-quinic acid decreased in S2/S1($\log_2$FC, $-1.03$) and increased in S3/S1($\log_2$FC, 1.33) and S4/S1($\log_2$FC, 2.97). Interestingly, the other metabolites identified by the pathway were all decreasing; the $\log_2$FC of S4/S1 was the minimum (Figure 8B).

## 4. Discussion

Seed germination is one of the most important growth and developmental processes in plants. It is an extremely complex process that is subject to the coordinated interaction of several genes, so it is only by studying the different germination stages of Chinese fir seeds that the molecular mechanisms of seed germination can be revealed.

In the study, high-throughput sequencing and metabolome analysis were used to evaluate differences in gene expression as well as metabolomics as a result of different developmental stages. Additionally, we also measured six physiological indicators related to seed germination. Different seed germination stages alter gene expression levels and metabolite content.

### 4.1. Dynamic Changes of Physiological Indexes during Seed Germination

SS is an important substance in seed germination and directly affects the consumption and degradation of starch within the seed and, thus, the supply of energy in embryonic growth. The seeds in this study reach a maximum at the S2 stage, probably because it is at this stage that the seeds need to break through the seed coat and therefore consume a large amount of nutrients to provide energy. SP and Pro contents play an important role in the regulation of osmotic potential in plant cells. In this study, SP and Pro contents reach their maximum values at stages S2 and S1, respectively, indicating that the strategies used to regulate the osmotic potential of plant cells at different stages of Chinese fir seed germination are different. MDA can affect the balance of the reactive oxygen metabolic system and is negatively correlated with the activities of SOD and POD. There are significant changes in MDA content in the germination of Chinese fir seed, and it plays an important role in the germination of the seed. SOD and POD activities are important components of the reactive oxygen system and are able to remove reactive oxygen species accumulated during seed germination in a timely manner. We were able to find a positive correlation between SOD and POD activities and reached a maximum at S2, indicating that reactive oxygen species are highest during the germination stage and that higher SOD and POD activities promote Chinese fir seed germination.

### 4.2. Changes of Transcriptome and Metabolome Levels during Seed Germination

Gene expression and metabolite content are closely related. Compared with S1, transcriptomic analysis indicated that the pathways significantly enriched in DEGs during early germination (S2) were 'Circadian rhythm—plant', 'Phenylpropanoid biosynthesis', 'Flavonoid biosynthesis', 'Plant hormone signal transduction', and 'Stilbenoid, diarylheptanoid and gingerol biosynthesis'. The pathways that were significantly enriched in late germination (S3, S4) were 'Photosynthesis', 'Flavonoid biosynthesis', 'Plant hormone signal transduction', and 'Phenylpropanoid biosynthesis'.

Metabolomic analysis showed that DAMs were enriched in 'ABC transporters', 'Flavonoid biosynthesis', 'Flavone and flavonol biosynthesis', and 'Phenylpropanoid biosynthesis' at an early stage (S2). At the late stage of germination (S3, S4), significantly enriched pathways were 'Flavonoid biosynthesis', 'Arachidonic acid metabolism', 'Biosynthesis of various plant secondary metabolites', 'Flavone and flavonol biosynthesis', and 'Phenylpropanoid biosynthesis'. We found that DEGs and DAMs were significantly enriched in phenylpropanoid biosynthesis and flavonoid biosynthesis during the germination of Chinese fir seeds. It has been shown that flavonoid and phenylpropanoid compounds

have important roles in plant stress, and the pathways of biosynthesis for these two are closely related [28,29]. Therefore, DEGs and metabolites in 'Flavonoid biosynthesis' and 'Phenylpropanoid biosynthesis' have important roles in the seed germination of Chinese fir.

### 4.3. The Key Pathways during Seed Germination

The secondary metabolites of plants play an important role in plant growth and development and resistance to stress [30,31]. The DEGs related to phenylpropanoid biosynthesis (CYP73A, COMT, 4CL, CCR, CAD, and PER) were significantly upregulated in Chinese fir seed germination. The gene 4CL (4-Coumarate-CoA ligase) is an important gene in phenylpropanoid biosynthesis, affects the biosynthesis of flavonoids, and plays important roles in plant physiology and stress [32,33]. Two 4CLs (4CL1 and 4CL2) were identified in our study; their expression was significantly upregulated in germination stages, and the expression of 4CL2 is higher than that of 4CL1, indicating that Chinese fir seed may influence the expressions of 4CL1 and 4CL2 to regulate the germination. Metabolites related to this pathway also increased significantly in late germination, such as caffeoylquinic acid and p-coumaroyl-CoA. These results showed that the expression of key genes related to biosynthesis and secondary metabolites was increased and promoted the accumulation of secondary metabolites, which had a positive effect on resistance to external environmental changes. Similar to our study, they also showed the importance of this pathway in seed germination [34,35]. The results explained that the pathway was important for seed germination of Chinese fir at the transcriptome and metabolome levels.

Flavonoids are common secondary metabolites that are widely distributed in plants [36,37]. Additionally, flavonoids have positive effects against environmental stress and multiple functions [38]. In this study, seed germination was affected by the transcription levels of key genes involved in the flavonoid biosynthesis pathway. Fifteen genes related to this pathway were detected to be differentially expressed in four development stages of Chinese fir seed. CHS is the first-committed enzyme and a key gene in the flavonoid biosynthesis pathway [28,39]. We identified two CHSs (CHS1 and CHS2), and they were both upregulated, with the maximum in S2. FLS and CHI related to flavonoid biosynthesis were important for plant growth [40,41] and upregulated in this study. However, it was worth noting that only CHI expression was decreased at the late stage of germination. Therefore, these DEGs of flavonoid biosynthesis needed to undergo complex regulation during Chinese fir seed germination. Interestingly, our metabolic analysis showed that the metabolites related to this pathway underwent significant changes, most of which decreased. Their content was highest in S1 and became less in the later developmental stages, such as apigenin, quercetin, and luteolin. However, kaempferol was accumulating in the late germination stage, while there was no difference in the S1 and S2 stages. These flavonoids may be the key substance affecting the seed germination of Chinese fir.

In addition, we noted that 'Plant hormone signal transduction' was also significantly enriched in the late stage of seed germination. The majority of genes associated with this pathway had upregulated expression, such as c186613.graph_c0, c237436.graph_c0, and c193003.graph_c0. Expression of the gene (c237436.graph_c0) consistently increased with seed germination, which could promote root formation. Additionally, the genes were closely related to the biosynthesis of ABA and GA. These results suggested that the expressions level of the genes affected hormone content and then seed germination.

### 5. Conclusions

In summary, we investigated the physiology, transcriptomic, and metabolic responses of Chinese fir seed during germination. Many genes and metabolites were changed in seed germination. Conjoint analysis of DEGs and DAMs showed that KEGG enrichment pathways were phenylpropanoid biosynthesis and flavonoid biosynthesis, and Chinese fir seed regulated the flavonoid and phenylpropanoid metabolites to affect germination. These results provided a theoretical basis for screening important genes and metabolites during Chinese fir seed germination.

**Supplementary Materials:** The following supporting information can be downloaded at: https:// www.mdpi.com/article/10.3390/f14040676/s1, Figures S1 and S2 are the GO enrichment analyses of S3/S1, S4/S1, respectively; Figures S3 and S4 are the KEGG analyses of S3/S1, S4/S1, respectively; Figure S5 is a qRT-PCR validation of six genes. Table S1 shows the statistics of RNA-seq sequencing data for 12 samples; Table S2 shows the primer sequences for the six genes and the internal reference genes; Table S3 shows the differentially expressed genes in the three comparison groups, and Tables S4 and S5 show the differential accumulation of metabolites for the three comparison groups in positive ion mode and negative ion mode, respectively.

**Author Contributions:** X.C. and G.Z. designed the study and drafted the article. Y.L. and S.W. helped with some of the experiments. Y.D. helped to collate the experimental data and assessed the presented data. Supervision: R.J. and All authors helped to edit the research paper. All authors have read and agreed to the published version of the manuscript.

**Funding:** This work was supported by Screening and pollution assessment of highly enriched plants for remediation of heavy metal contaminated soil.

**Data Availability Statement:** Transcriptome raw data has been uploaded to NCBI under project number PRJNA898903.

**Conflicts of Interest:** The authors declare no conflict of interest.

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
