# Peer review of "Integrative Analysis of the Transcriptome and Metabolome Reveals the Mechanism of Chinese Fir Seed Germination"

_forests, doi:10.3390/f14040676_

Round 1

Reviewer 1 Report

There are a number of minor comments to the manuscript.

Аbstract. DEG, DAM, etc. should be defined at the first mention.

L. 54-56. ”However, changes at the level of transcriptional and metabolism involved in the germination process of Chinese fir seed have not been reported”. References to the transcriptome and metabolome analysis in germinating seeds of other forest trees should be given before this statement.

“References must be numbered in order of appearance in the text and should be placed in square brackets; for example [1], [1–3]” (Instructions for Authors).

Subsection 2.4. Here you need to specify the list of primers (Suppl. Table 2) and the reference gene.

Fig. 2. What do bars mean? SE or SD?

Suppl. Material must be listed after the Conclusion section.

Ref. 1, 6, 7, 9, etc. Page numbers are missing.

English should be edited.

Author Response

Dear reviewers,

Thank you for your letter. We were pleased to know that our work was rated as potentially acceptable for publication in Journal, subject to adequate revision. We thank the reviewers for their time and effort in reviewing the previous version of the manuscript. Their suggestions have enabled us to improve our work. Based on the instructions provided in your letter, we uploaded the file of the revised manuscript.

Appended to this letter is our point-by-point response to the comments raised by the reviewers. The comments are reproduced and our responses are given directly afterward in a different color (red).

We would like also to thank you for allowing us to resubmit a revised copy of the manuscript. We hope that the revised manuscript is accepted for publication in the Journal of Forests.

Thanks again!

Point 1: Аbstract. DEG, DAM, etc. should be defined at the first mention.

Response 1: DEG,DAM,GO and KEGG were defind in abstract, and these were added to the revised manuscript.

Point 2:L. 54-56. ”However, changes at the level of transcriptional and metabolism involved in the germination process of Chinese fir seed have not been reported”. References to the transcriptome and metabolome analysis in germinating seeds of other forest trees should be given before this statement.

Response 2: We have not found any transcriptome or metabolome related studies on seed germination in forest trees, and such studies have focused on fiddler crops and cash crops.

Point 3: “References must be numbered in order of appearance in the text and should be placed in square brackets; for example [1], [1–3]” (Instructions for Authors).

Response 3: The formatting of the references has been adjusted in the revised manuscript.

Point4:Subsection 2.4. Here you need to specify the list of primers (Suppl. Table 2) and the reference gene.

Response 4: “The six genes were used for the qRT-PCR analysis (Supplement Table2).”This was added to Subsection 2.4.

Point5: Fig. 2. What do bars mean? SE or SD?

Response 5: “And all values are mean ± standard error, different letters above the bars indicate significant differences.” This phrase is added to Figure 2.

Point 6:Suppl. Material must be listed after the Conclusion section.

Response 6:” Transcriptome raw data has been uploaded to NCBI, project number PRJNA898903; Metabolome raw data uploaded to MetaboLights, project number MTBLS7427.

FigureS1, FigureS2 are the GO enrichment analysis of S3/S1,S4/S1 respectively; FigureS3,FigureS4 are the KEGG analysis of S3/S1,S4/S1 respectively; FigureS5 is a qRT-PCR validation of six genes.

Table1 shows the statistics of RNA-seq sequencing data for 12 samples; Table2 shows the primer sequences for the six genes and the internal reference genes; Table3 shows the differentially expressed genes in the three comparison groups; Table4 and Table5 show the differential accumulation of metabolites for the three comparison groups in positive and negative ion mode respectively.” These were listed after the conclusion.

Point 7:Ref. 1, 6, 7, 9, etc. Page numbers are missing.

Response 7: The missing page numbers were added to the revised manuscript.

Point 8:English should be edited.

Response 8: I have done my best to check the language of the manuscript and make some minor changes, thank you for your interest.

Reviewer 2 Report

In this study, the authors conducted an intensive analysis of the transcriptome and metabolome to reveal the seed germination mechanism in Chinese fir. The experiments were nicely done. However, the materials and methods for RNA sequencing were poorly described. The quality of many figures is very poor. Raw data for transcriptome and metabolome should be deposited in public databases, and then the accession numbers for the raw data should be provided in the manuscript. I suggest this manuscript for major revision. My comments are as follows:

The abstract appears to be fine. Please spell out abbreviations for differentially expressed genes (DEGs) and differentially accumulated metabolites (DAMs).

L74-76: Can you provide the timeline for stages S1 to S4 of seed germination?

L76: FIGURE 1 -> Figure 1

Section 2.3: Please describe the methods used to calculate the expression of genes using raw data. The reference for the mapping should be provided, and the names of databases should be spelled out. The accession dates for the databases should be provided. For GO enrichment analysis, the source of the reference genome should be provided.

Section 2.5: Spell out abbreviations such as fold change (FC), VIP, and databases used for the metabolite analysis. L132: P-value -> "P" should be italicized.

In Figure 2, the font size should be increased, and the graphs should be separated from each other. I suggest using a different color bar for each stage.

In Figure 3, the font sizes can be increased. Please indicate the number of DEGs in the graph in panel A. Panel C should be magnified, and the quality of panels C and D should be improved.

L214: Metabolomic analyze -> Metabolome analysis. For the metabolome analysis, it might be good to include a table showing the representative metabolites that were significantly changed during seed germination.

In Figure 5, the font sizes should be increased, and the quality of images is poor. Please replace Figure 5 with high-quality images.

Panel A should read "(A) DAMs of seeds."

The quality of Figures 6, 7, and 8 is poor and small. Please revise them.

Author Response

Dear reviewers

Thank you for your letter. We were pleased to know that our work was rated as potentially acceptable for publication in Journal, subject to adequate revision. We thank the reviewers for their time and effort in reviewing the previous version of the manuscript. Their suggestions have enabled us to improve our work. Based on the instructions provided in your letter, we uploaded the file of the revised manuscript.

Appended to this letter is our point-by-point response to the comments raised by the reviewers. The comments are reproduced and our responses are given directly afterward in a different color (red).

We would like also to thank you for allowing us to resubmit a revised copy of the manuscript. We hope that the revised manuscript is accepted for publication in the Journal of Forests.

Thanks again!

Point 1: The abstract appears to be fine. Please spell out abbreviations for differentially expressed genes (DEGs) and differentially accumulated metabolites (DAMs).

Response 1: DEG,DAM,GO and KEGG were defind in abstract, and these were added to the revised manuscript.

Point 2:L74-76: Can you provide the timeline for stages S1 to S4 of seed germination?

Response 2: “In addition, the four germination stages were sampled on the first, third, fifth and eighth days of the germination experiment.” This was added to L75-76.

Point 3:L76: FIGURE 1 -> Figure 1

Response 3: We have changed FIGURE1 to Figure1 in the 75 line.

Point 4:Section 2.3: Please describe the methods used to calculate the expression of genes using raw data. The reference for the mapping should be provided, and the names of databases should be spelled out. The accession dates for the databases should be provided. For GO enrichment analysis, the source of the reference genome should be provided.

Response 4: “The sequenced Reads were compared with the Unigene library using Bowtie, and the expression levels were estimated based on the results of the comparison in combination with RSEM. The FPKM values were used to express the expression abundance of the corresponding Unigene.” “As the material for this experiment was reference-free, GO functional annotation of genes from this species was required prior to enrichment analysis. The annotated GO was then used as the background gene set for GO enrichment analysis.” These were added to Section 2.3.

Point 5:Section 2.5: Spell out abbreviations such as fold change (FC), VIP, and databases used for the metabolite analysis. L132: P-value -> "P" should be italicized.

Response 5: FC,VIP, P-value and databases were supplemented in L139-144.

Point 6: In Figure 2, the font size should be increased, and the graphs should be separated from each other. I suggest using a different color bar for each stage.

 Response 6: Figure 2 has a larger font size and a different colour for each stage in revised manuscript.

Point 7:In Figure 3, the font sizes can be increased. Please indicate the number of DEGs in the graph in panel A. Panel C should be magnified, and the quality of panels C and D should be improved.

 Response 7: We have adapted and modified Figure 3 as required.

Point 8:L214: Metabolomic analyze -> Metabolome analysis. For the metabolome analysis, it might be good to include a table showing the representative metabolites that were significantly changed during seed germination.

 Response 8: We have changed Metabolomic analyze to Metabolome analysis in L234. And the table of representative metabolites were included inSupplemnet Table6.

Point 9:In Figure 5, the font sizes should be increased, and the quality of images is poor. Please replace Figure 5 with high-quality images.

Panel A should read "(A) DAMs of seeds."

 Response 9: Figure 5 has been adjusted and modified.

Point 10:The quality of Figures 6, 7, and 8 is poor and small. Please revise them

Response 10: We have revised Figures 6, 7 and 8 in revised manuscript.

Round 2

Reviewer 2 Report

The authors have properly revised their manuscript according to the reviewers' comments.

Based on my assessment, I recommend that this manuscript be accepted for publication.